# Mental health burden among females living with HIV and AIDS in sub-Saharan Africa: A systematic review

Dorothy Serwaa Boakye[1,2]*, Mawuko Setordzi[2,3], Gladys Dzansi[2],
Samuel Adjorlolo[2,4]

**1** Department of Health Administration and Education, University of Education, Winneba, Ghana, **2** School of
Nursing and Midwifery, College of Health Sciences, University of Ghana, Legon, Accra, Ghana,
**3** Department of Nursing, Presbyterian Nursing, and Midwifery Training College, Dormaa Ahenkro, Bono
Region, Ghana, **4** Research and Grant Institute of Ghana, Legon, Ghana

* dsboakye@uew.edu.gh

journal.pgph.0002767

Science, UNITED STATES

**Data Availability Statement:** All data underlying
the findings in this manuscript has been submitted
as Supporting information.

## Abstract

Mental health problems, particularly depression and anxiety, are common in women and
young girls living with HIV/ AIDS particularly in low- and middle-income (LMICs) countries
where women's vulnerability to psychiatric symptoms is heightened due to the prevalent
intersectional stressors such as stigma and intimate partner violence. However, no syn-
thesized evidence exists on the mental health burden of females living with HIV/AIDS
(FLWHA) in Africa. This systematic review aimed to synthesize the current evidence on
the mental health burden among FLWHA in sub-Saharan Africa. A systematic literature
review of articles published from 2013–2023 was conducted using the Preferred Report-
ing Items for Systematic Reviews and Meta-Analyses guidelines (PRISMA). Five elec-
tronic databases; PubMed, MEDLINE with full text, Scopus, Academic Search Complete,
and Health Source: Nursing Academic Edition were searched for articles published in
English. Nineteen articles (15 quantitative, 3 qualitative, and 1 case study) from over 7
African countries met the inclusion criteria. The majority of the studies' quality was deter-
mined to be moderate. The prevalence of depression ranged from 5.9 to 61% and anxiety
from 28.9 to 61%. Mental health burden was a logical outcome of HIV diagnosis. Predic-
tors of mental health outcomes in the context of HIV/AIDS were identified as intimate part-
ner violence (IPV), stigma, childhood traumas, sexual abuse, poverty, unemployment,
and social isolation. Social support and resilience were identified as protective factors
against mental illness in FLWHA. Mental illness had a deleterious effect on viral suppres-
sion rates among FLWHA, resulting in delayed initiation of antiretroviral therapy treatment
and increased mortality but had no impact on immune reconstitution in the face of ART
adherence. Given the high prevalence rates of depression and anxiety and their relation-
ship with HIV progression, it is crucial that mental health care services are integrated into
routine HIV care.

**Funding:** The authors received no specific funding for this work.

**Competing interests:** The authors have declared that no competing interests exist.

## 1. Introduction

Since the beginning of the HIV/AIDS epidemic, women and girls have been most infected and affected. The joint United Nations Program on HIV/AIDS reports show that globally, more than half (20.2 million) of the 37.7 million people living with HIV and AIDS are females [1]. The numbers recorded in sub-Saharan Africa are even more devastating. Approximately, two-thirds of the estimated 25.6 million people currently living with HIV and AIDS in sub-Saharan Africa are females [1, 2].

Mental health conditions affect females living with HIV and AIDS (FLWHA) at substantially higher rates compared with men living with HIV and AIDS (MLWHA) and HIV-negative women [3–6]. Thus, the established gender disparity in mental health disorders such as depression and anxiety is disproportionately increased in the context of HIV and AIDS [7]. Given the psychological distress associated with managing and coping with the illness, a diagnosis with HIV/AIDS may cause or worsen pre-existing mental illness symptomatology in women [4, 8].

FLWHA's relative vulnerability to mental health conditions may be due to several intersectional stressors such as unemployment, financial stressors, decreased psychological resilience, limited social support, and increased susceptibility to neuropsychiatric side effects of some antiretroviral therapies such as zidovudine and abacavir [5, 9, 10]. Additionally, FLWHA in low- and middle-income countries have higher rates of intimate partner violence, stigma, discrimination, and social prejudice [10, 11, 12] as well as increased psychosocial stress with HIV status disclosure and sexual and reproductive health management [12, 13].

Higher rates of depressive and anxiety symptoms have been reported amongst FLWHA in several studies [3–7]. For instance, 82% of survey respondents in a study of 832 FLWHA from 94 countries reported having symptoms of depression, and participants indicated that their mental health concerns rose 3.5-fold after receiving an HIV diagnosis [14]. Furthermore, 28.9% of 357 FLWHA ages 18 years and above in an Ethiopian study experienced moderate to high levels of anxiety, as measured by a score of at least 11 on the Hospital Anxiety and Depression Scale (HADS-A) [6]. As depressive and anxiety disorders are frequently underdiagnosed among persons living with HIV and AIDS (PLWHA), these rates are likely to be higher than reported [15].

Though a substantial burden of HIV associated depression exist in sub-Saharan Africa [16], the majority of the mental health interventions for this population have been developed and evaluated in resource-rich settings and with a limited focus on FLWHA. Along with this limitation even exist a major gap as there is a lack of systematic reviews that synthesize the available research evidence on mental health illness among FLWHA in Africa to direct policymakers to implement cost-effective and evidenced-based gender-specific interventions that are peculiar to our cultural context as Africans. The overall goal of this review was to synthesize the current existing knowledge on the specific mental health issues faced by FLWHA in Africa. Our study may have implications for the United Nations 95-95-95 treatment target by 2025 since the evidence shows FLWHA with untreated mental illness are less likely to adhere to their antiretroviral medication (ART) and more likely to experience virologic failure and HIV progression [17].

### 1.1 Aim of the review

The primary goal of this review was to synthesize the current understanding of the specific mental health problems among FLWHA in Africa and describe the relationship between HIV/AIDS and mental health.

### 1.2 Objectives of the review

1. Determine the prevalence of depression and anxiety among FLWHA.

2. Identify the risks, and protective factors of mental health problems among FLWHA

3. Describe the impact of HIV/AIDS on the mental health burden of FLWHA.

4. Describe the impact of mental health burden on HIV progression among FLWHA

### 1.3 Review question

The main question that guided this systematic review methodology was: "Does HIV AND AIDS status negatively impact on the mental health of FLWHA?" (Table 1).

## 2. Methods

This systematic review adheres to the Preferred Reporting Items for Systematic Reviews and Meta Analysis (PRISMA) guidelines [18] (S1 Checklist). Before the commencement of the review, we conducted an initial search on Prospero and Cochrane Library using the main concepts of the topic in Table 2 to ascertain if the systematic review has been done already to avoid unintentional duplication and redundancy. We noticed that systematic reviews were conducted in the general population but not specific to FLWHA. Again, this search gave us a fair idea about the availability of literature on the topic. During the building process for the search terms, relevant synonyms were identified for each of the main concepts as shown in Table 2. All the key concepts and their synonyms were combined to construct a comprehensive search strategy for the identified databases. The protocol for this systematic review was registered via the PROSPERO database (CRD42023448947). The study focused on studies on mental health burden in females living with HIV/AIDS in Africa.

### 2.1 Inclusion and exclusion criteria

Articles were included in the review if they focused on mental health problems in females living with HIV/AIDS between 17 and 76 years and published in English in the last ten years from 2013 to April 2023. Articles focused on pregnant and post-natal women were excluded because 1) systematic reviews exist in the literature in Africa on this population and 2) pregnancy and its associated hormonal imbalance are considered covariates to mental health burden in FLWHA [19, 20]. The details of the criteria for inclusion and exclusion have been described in detail in Tables 3 and 4.

### 2.2 The electronic database search

For the electronic search, five (5) major databases were selected. A systematic search of electronic databases using pre-defined search terms, supplemented by hand-searching, was undertaken in April 2023. These databases included: PubMed, Scopus, MEDLINE with full text,

**Table 1. Review question in PEO format.**

| PEO format | Concepts |
| --- | --- |
| **P**-Population- | Females |
| **E**-Exposure - | HIV/AIDS |
| **O**-Outcome- | Mental Health burden |

**Table 2.  Search strategy for Pubmed using Boolean Operators (OR & AND).**

| PEO Format | Combined Text words and MeSH Words |
|---|---|
| **P- population** | "Female"[Mesh] OR Female*[tw] OR Woman [tw] OR Women [tw] |
| **E- exposure** | "HIV"[Mesh] OR HIV/AIDS [tw] OR "HIV-positive" [tw] OR "HIV and AIDS" [tw] OR "Acquiredimmunodeficiency syndrome" [tw] OR "Human immunodeficiency virus" |
| **O- outcome** | "Mental Health"[Mesh] OR "Mental health"[tw] OR "Mental wellbeing"[tw] OR "Psychological wellbeing"[tw] OR Depress*[tw] OR Anxiety[tw] OR "Mental disorder*"[tw] OR "Psychological disorder*"[tw] OR "Mental illness*"[tw] OR "Psychological distress"[tw] OR "Mental distress"[tw] OR "Emotional distress"[tw] |
|  | "Africa"[Mesh] OR Africa [tw] OR "Sub Saharan Africa" [tw] |

**Table 3.**

| Inclusion criteria |
|---|
| 1 Mental health in females between 17 and 76 years living with HIV/AIDS |
| 2 Scholarly articles on depression and anxiety among females living with HIV/AIDS |
| 3 Studies that employed quantitative, or qualitative research designs |
| 4 Scholarly articles published in English |
| 5 Scholarly articles published within the last ten years (2013–2023) |
| 6 Articles accessible as full text |

**Table 4.**

| Exclusion criteria |
|---|
| 1 Studies describing mental health burden in pregnant females living with HIV/AIDS |
| 2 Studies describing mental health burden in post-natal women living with HIV/AIDS |
| 3 Studies specifically focused on mental health interventions among FLWHA |
| 4 Review articles |

Academic Search Complete, and Health Source: Nursing Academic Edition. Additionally, the reviewers searched Google Scholar engine, WHO website, UNAIDS website, Ghana's Mental Health Authority website and approved institutional websites (Ghana AIDS Commission, Ghana Health Service) reporting findings and statistics on HIV/AIDS and mental illness for both grey and published literature. Also, the reference lists of the papers that were chosen for inclusion were used as a snowball for additional studies. Two reviewers (DSB & MS) conducted the final electronic search independently in the selected databases on 24 April 2023 with the help of the designed search strategy.

## 2.3 Conducting the electronic search

First, articles retrieved from the various relevant electronic databases and the other sources were title screened by the reviewers (DSB & MS) to determine whether they were relevant for inclusion. Second, the abstracts of the respective articles were also screened to ascertain whether their findings will contribute meaningfully to the research questions of the review. Third, the full text of the respective articles was screened and those that were found significant were added. At this stage reasons for exclusion of articles were provided (see Fig 1). Duplicates

**PRISMA 2020 flow diagram for new systematic reviews which included searches of databases, registers and other sources**

"Fig 1"

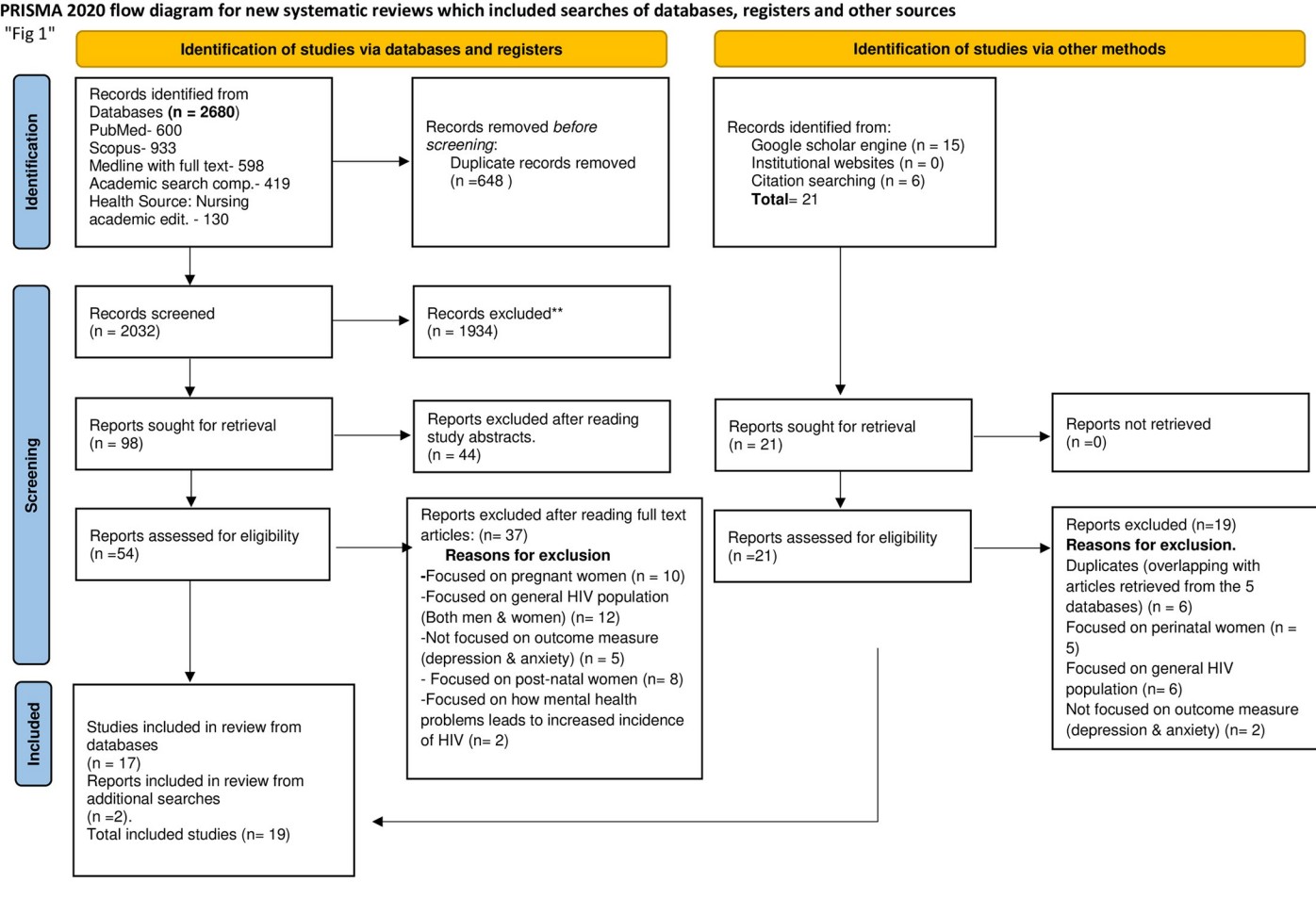

**Fig 1. Updated 2020 PRISMA flow diagram.**

were removed manually during the selection process and also by the use of Rayyan, a systematic review software built with artificial intelligence (AI).

## 2.4 Selection of studies and data extraction

All titles and abstracts identified through electronic database searching and hand searching were examined for relevance. Full text articles of any potentially relevant studies were obtained and assessed for inclusion based on the above inclusion criteria. Where uncertainty arose within the examination process, full text articles were subjected to further scrutiny by the inclusion of a third reviewer (S.A). S.A who doubled as project supervisor and reviewer, also assisted with validation and review of a subsection of identified titles and abstracts. Less than 5% discrepancies arose between the authors (DBS & MS) and the third reviewer. These articles were not included in the review. Data from studies that met the inclusion criteria were extracted using an extraction format adapted from a recent systematic review [21]. Data extraction was carried out by the first and second authors (DSB & MS) and, scrutinized by the fourth author (S.A). Information extracted from relevant articles included publication details such as simple size and its characteristics, country, measures used, authors and year of publication. The objectives of this review guided the extraction of the major findings in the included studies.

## 2.5 Data extraction from included studies

Using the adapted data extraction format, relevant data was retrieved from each included study for the analysis. The author and year of publication, the title of publication, the country where the study was done, the sample size and characteristics, measure used, prevalence, risks/ correlates, protective factors, impact of HIV/AIDS on mental health, and the impact of mental health on HIV progression were extracted from each including. Special attention was paid to the methods of recruitment/selection of participants, data collection approach and data analysis done for each of the studies. Data were extracted into tables with pre-specified categories as stated previously. For quantitative studies, we collected data on prevalence of depression and anxiety, associated risk factors, impact of HIV/AIDS on mental health and impact of mental health burden on HIV progression. Main findings identified by the authors were summarised. For the qualitative studies, the authors collected information about the experiences of depression/anxiety, risk and coping. Data was extracted and synthesized collaboratively by two authors (MS & DSB). The fourth author (SA) validated and made suggestions on the themes. Some were modified and subsequently included in the review (Table 5).

## 2.6 Assessment of quality and risk of bias of included studies

The quality of the included studies were assessed using Joanna Briggs Institute (JBI) Critical Appraisal Tools [22]. This group of tools comprised checklists adopted for a range of study designs that includes a 9-item checklist for studies reporting prevalence data [23], an 8-item checklist for case reports [24] and a 10 item checklist for qualitative studies [25]. The checklists pose a series of questions regarding presence of specific information with response options "yes = 1", "no = 0", "unclear = 0", and "not applicable = not scored" to guide appraisers toward conclusions about the methodological quality of a study or analysis. (See Tables 6–8). The 9-item checklist was awarded an overall score of 9 with scores of 1–3, 4–6 and 7–9 indicating low, moderate and high quality. The 8-item checklist was awarded a score up to 8 with scores of 1–3, 4–6 and 7–8 indicating low, moderate and high quality. The overall score of the 10-item checklist was 10 with scores of 1–3, 4–6 and 7–10 indicating low, moderate and high quality. The response score of each study were summed up to determine if the study was of low, moderate or high quality.

## 3. Results

### 3.1 Characteristics of the included studies

The search and screening processes is illustrated in Fig 1. A total of 19 articles were included in the review. All of these studies were published between 2014 and 2023. Out of the 19 articles, 15 were quantitative, 3 qualitative and 1 case study. The participants were aged between 18 and 76 years. In all 12,299 (adolescents, adults and the aged) made up the entire sample size for the nineteen (19) studies. The major findings in the included studies have been summarized in Table 5 (summary of data extracted from included studies). Most of the studies (n = 9, 52.6%) were conducted in South Africa, followed by Kenya (n = 2, 11.1%), Uganda (n = 2, 11.1%) whilst the remaining were conducted in Malawi (n = 1,5.5%), Tanzania (n = 1, 5.5%), Ethiopia (n = 1. 5.5%) Nigeria (n = 1, 5.5%), and parts of sub-Saharan Africa (n = 1, 5.5%). One study was conducted both in Zambia and Uganda (n = 1, 5.5%). For this review, the themes that appeared in at least two papers were kept.

**Table 5. Summary of data extracted.**

| NO | Author/Year | Method/Design | Country | Sample size & Sample characteristic | Measures used for MHB | Risk for MHB in FLWHA | Protective factors of MHBs | HIV/AIDS impact on MHB | MHB impact on HIV/AIDS Progression | Prevalence |
|----|-------------|---------------|---------|-------------------------------------|------------------------|------------------------|----------------------------|------------------------|-------------------------------------|-----------|
| 1 | Casale et al., (2015) | Quantitative, cross-sectional study | South Africa | 2,199 female primary caregivers, 18 years or older | Centre for Epidemiologic Studies Depression Scale (CES-D) | Living in the rural site. Worse socio-economic status. HIV/AIDS status | | HIV/AIDS appeared to be a greater risk factor for depression than other illness. Depression was considerably higher among HIV-positive caregivers. than other ill caregivers and healthy caregivers | | The prevalence of depression was 56.3%) |
| 2 | Kamen et al., (2015) | Quantitative, cross-sectional study | Malawi | 59 HIV-positive women within 17–46 years. | Seven-item scale related to symptoms of anxiety and two-item scale related to worry (Emotional distress scale) | Living in homes without electricity and water. Poor living conditions. HIV stigma was also a significant predictor of emotional distress | | | | The prevalence of anxiety (emotional distress) 53% |
| 3 | Yemeke et al., (2020) | Quantitative, cross-sectional survey | South Africa | 70 female patients who were enrolled in HIV care at the study clinic. Were 18 years and above. | The Patient Health Questionnaire–2 (PHQ-2) screener | Sexual abuse Physical IPV | | | | Prevalence of depression 36%, |
| 4 | Tsai et al., (2016) | Longitudinal study | Uganda | 173 HIV-positive women in rural Uganda | Hopkins Symptom Checklist (HSCL-D; that was modified for the local context by Bolton and Ndogoni (2000) | Victimization Forced sex | | | | Prevalence of depression 33% |
| 5 | Rossouw et al., (2021) | Quantitative, cross-sectional study | South Africa | 410 cisgender FSW who were aged 18 years and above | The Patient Health Questionnaire-9 (PHQ-9) instrument | Living with people other than direct family members such as friends, boyfriends, or colleagues Staying alone Being homeless, High social stigma Sexual violence Physical violence | | HIV positive status had no relationship with depression | | Prevalence of depression 28.8%. |

*(Continued)*

**Table 5.** (*Continued*)

| NO | Author/ Year | Method/ Design | Country | Sample size & Sample characteristic | Measures used for MHB | Risk for MHB in FLWHA | Protective factors of MHBs | HIV/AIDS impact on MHB | MHB impact on HIV/AIDS Progression | Prevalence |
|---|---|---|---|---|---|---|---|---|---|---|
| 6 | Williams et al., (2020) | Qualitative, exploratory study. | South Africa | 27 young women aged 18 and 35 above. living with HIV and on ART treatment. | Center for Epidemiologic Studies Depression Scale (CES-D ≥ 16) | Complex and traumatic life experiences, intimate partner violence, Sexual assault, self-stigma, loss of self-esteem, experiences of stigma and discrimination, loss of life partners/ husbands, and other family member | | | | |
| 7 | Opiyo et al., (2016) | Case study | Kenya | A 42-year-old unmarried woman HIV positive | The Beck Depression index | Stigma, gender-based violence, loss of livelihood | | | | |
| 8 | Spies and Seedat (2014) | Quantitative, cross-sectional study | South Africa | 95 HIV positive women aged between 18 and 63years Mean age-33.6 | Center for Epidemiologic Studies Depression Scale (CES-D) | childhood trauma and post-traumatic stress symptoms | Resilience was significantly associated with decreased depressive symptoms | | | Prevalence of current depression6.5% and recurrent depression 8.4% |
| 9 | Kako et al., (2016) | Qualitative | Kenya | 54 HIV positive women between the ages 18 and 69 years. Mean age was 37. | Semi structured interview | Worry, feelings of hopelessness, uncertainty, HIV related stigma. | | HIV diagnosis led to intense psychological distress | | |
| 10 | Govender et al., (2022) | Quantitative, cross-sectional study. | South Africa | 2955 young and older women who were HIV positive and between the ages 15–49 years. Mean age 21.8 | Semi -structured interview guide | Age | | | higher depression scores were associated with a reduced likelihood of being virally suppressed in younger women but not in older women. Higher depression scores in both ages were less likely to have utilized HIV care services and initiate ART | |

(*Continued*)

**Table 5.** (Continued)

| NO | Author/ Year | Method/ Design | Country | Sample size & Sample characteristic | Measures used for MHB | Risk for MHB in FLWHA | Protective factors of MHBs | HIV/AIDS impact on MHB | MHB impact on HIV/AIDS Progression | Prevalence |
|---|---|---|---|---|---|---|---|---|---|---|
| 11 | Burgess & Campell (2014) | Qualitative study | South Africa | 19 HIV positive women who were aged between 18 and 76 years. | semi structured interviews | Poverty Violence HIV/ AIDS Women highlighted family conflicts (particularly abandonment by men), community-level violence, poverty and HIV/AIDS as drivers of distress. | Resilience, psychological re-framing of negative situations, mobilisation of emotional and financial support from inter-personal networks, churches and HIV support groups. They also sought expert advice from traditional healers, medical services or social workers, | HIV did not emerge as an independent driver of emotional distress | | |
| 12 | Rael et al., (2021). | Quantitative Cross-sectional | Parts of sub-Saharan Africa | 190 women old than 18 years old | Eight-item, depression-focused subset of the Hopkins Symptom Checklist (HSCL-25) | | | | clinical depression was positively associated with high HIV VL. | The prevalence of depression was 38.7% |
| 13 | Ortblad et al., (2020). | Quasi-experimental design | Uganda and Zambia | 1925 participants, 960 in Uganda and 965 in Zambia. | Patient Health Questionnaire-9 item (PHQ-9) depression scale | | | Knowledge of HIV-positive status was significantly associated with a decrease in depressive symptoms in Uganda and in Zambia. | | The prevalence of depression 43.2% |
| 14 | Sudfeld et al., (2017) | Prospective cohort study | Tanzania. | 1,487 women who initiated ART in Dares Salaam. | Tanzanian adapted and validated version of the Hopkins Symptom Checklist (HSCL-25). | Stigma | Social support | | depression at ART initiation was associated with an increased risk of mortality and incidence of severe anemia. Depression was not significantly associated with trajectory of CD4 T-cell reconstitution or the risk of immunologic failure. | Prevalence of depression was 57.8% |

(*Continued*)

**Table 5.** (Continued)

| NO | Author/ Year | Method/ Design | Country | Sample size & Sample characteristic | Measures used for MHB | Risk for MHB in FLWHA | Protective factors of MHBs | HIV/AIDS impact on MHB | MHB impact on HIV/AIDS Progression | Prevalence |
|---|---|---|---|---|---|---|---|---|---|---|
| 15 | Familiar et al., (2016). | Quantitative, Cross-sectional | Uganda | 288 HIV-infected women | Hopkins Symptom Checklist (HSCL-25) | Lower wealth lack of community support, Poverty | family support, social support | | | Depression or anxiety symptoms were reported by 61% of participants |
| 16 | Spies et al., (2018). | Quantitative, Longitudinal study | South Africa | 148 HIV-infected and uninfected women 68 infected 80 uninfected | The Center for Epidemiologic Studies Depression Scale (CES-D) | Post-traumatic stress disorder, childhood trauma | | Higher rates of depressive symptomatology was persistent among HIV infected women compared to uninfected women | | The prevalence of depression was 20% |
| 17 | Abiodun et al., (2022) | Cross-sectional study | Nigeria | 458 adult women accessing HIV care | The 21-item Beck Depression Inventory 11 (BDI-11) | Social isolation | Social support | | | Prevalence of depression was 5.9% |
| 18 | Yousuf et al (2020) | Quantitativ e, cross sectional study | Ethiopia | 357 women aged 18 years and above attending HIV treatment service | The Hospital Anxiety and Depression Scale (HADS) | Gender No formal education Employment status. HIV/AIDS Stigma Divorced, Opportunistic infection, | Perceived social support | Women with symptomatic HIV clinical stage IV were more likely to get anxiety and clinical stage III more likely to get depression | | Prevalence of depression was 32.5% and anxiety was 28.9%, respectively, |
| 19 | Bhardway et al. (2023) | Quantitative cross sectional study. | South Africa | 1,384 female sex workers with a median age of 31 years | Patient Health Questionnaire-9 item (PHQ-9) depression scale | Sexual violence, physical violence, internalizing stigma, illicit drug use | Social support | depression was associated with a 24% increase in prevalence of unsuppressed viral load | | 33.2% of enrolled women screened positive for depression |

### 3.2 Prevalence of depressive symptoms in FLWHA

Data on the prevalence of depression were published by Thirteen (13) studies (68.4%) [6, 26–34, 36, 40–42]. According to the data that were retrieved, the prevalence of depressive symptoms varies from 5.9%, among HIV-positive women in Nigeria [34] to 61% among HIV-positive women in Uganda, [33] demonstrating the lowest to the highest depression symptomatology at different study locations. High prevalence rates were reported by two studies conducted in South Africa 56.3% [27] and Uganda 61%[33]. Relatively lower rates of depression were reported in two studies. Thus 5.9% in Nigeria [34] and 6.5% in South Africa [32]. In a case study conducted in Kenya on a 42 year old unmarried FLWHA, she indicated fifteen (15) symptoms corresponding to mild depression on the Beck Depression Index [35].

### 3.3 Prevalence of anxiety symptoms in FLWHA

The prevalence of anxiety symptomatology was reported in three (3) studies [6, 28, 36] representing 15.8%. The anxiety prevalence ranges from 28.9% among FLWHA in Ethiopia [6] to 61% among FLWHA in Uganda [36]. The reported anxiety prevalence in these three studies in terms of severity ranges from 61.0% [36] in Uganda, 53.0% [28] in Malawi and 28.9%, [6] in

**Table 6. JBI critical appraisal checklist for studies reporting prevalence data.**

| AUTHOR'S NAME/YEAR | Was the sample frame appropriate to address the target population? | Were study participants sampled in an appropriate way? | Was the sample size adequate? | Were the study subjects and the setting described in detail? | Was the data analysis conducted with sufficient coverage of the identified sample? | Were valid methods used for the identification of the condition? | Was the condition measured in a standard, reliable way for all participants? | Was the appropriate statistical analysis? | Was the response rate adequate, and if not, was the low response rate managed appropriately? | Overall score | Remarks |
|---|---|---|---|---|---|---|---|---|---|---|---|
| Casale et al (2015) | Yes | Yes | Yes | Yes | Yes | Yes | Yes | Yes | Yes | 9 | High Quality |
| Kamen et al (2015) | Yes | Unclear | No | Yes | Yes | Unclear | Unclear | Yes | No | 4 | Moderate quality |
| Yemeke et al (2020) | Yes | Unclear | No | Yes | Yes | Yes | Unclear | Yes | Unclear | 5 | Moderate quality |
| Tsai et al (2016) | Yes | Unclear | No | Yes | Yes | Yes | Yes | Yes | Unclear | 6 | Moderate quality |
| Rossouw et al (2021) | Yes | No | Unclear | Yes | Yes | Yes | Yes | Yes | Yes | 7 | High Quality |
| Spies and Seedat (2014) | Yes | No | No | Yes | Yes | Yes | Yes | Yes | Unclear | 6 | Moderate quality |
| Rael et al (2021) | Yes | Yes | No | Yes | No | Yes | Yes | Yes | Unclear | 6 | Moderate quality |
| Govender et al (2021) | Yes | Yes | Yes | Yes | Yes | Yes | Yes | Yes | Yes | 9 | High Quality |
| Ortblad et al (2020) | Yes | Yes | No | Yes | Yes | Yes | Yes | Yes | Unclear | 7 | High Quality |
| Sudfeld et al (2017) | Yes | Yes | Yes | No | Yes | Yes | Yes | Yes | Yes | 8 | High Quality |
| Familiar et al (2015) | Yes | No | Unclear | No | Yes | Yes | Yes | Yes | Yes | 6 | Moderate quality |
| Spies et al (2018) | Yes | Unclear | No | No | Yes | Yes | Yes | Yes | Yes | 6 | Moderate quality |
| Abiodun et al (2021) | Yes | Yes | Yes | No | Yes | Yes | Unclear | No | Yes | 5 | Moderate quality |
| Yousuf et al (2020) | Unclear | Yes | Yes | Yes | Yes | Yes | Unclear | Yes | Yes | 7 | High Quality |
| Bhardwaj et al (2023) | Yes | No | Yes | Yes | Yes | Yes | Yes | Yes | Yes | 8 | High Quality |

**Table 7. JBI critical appraisal checklist for qualitative research.**

| AUTHOR'S NAME | Is there congruity between the stated philosophical perspective and the research methodology? | Is there congruity between the research methodology and the research question or objectives? | Is there a congruity between the research methodology and the methods used to collect data? | Is there a congruity between the research methodology and the representation and analysis of data? | Is there a congruity between the research methodology and the interpretation of the results? | Is there a statement locating the reasearcher culturally or theoritically? | Is the influence of the researcher on the research and vice-versa addressed? | Are participants, and their voices, adequately represented? | Is the research ethical according to current criteria or, for recent studies, and is there evidence of ethical approval by an appropriate body? | Do the conclusions drawn in the research report flow from the analysis, or interpretation, of the data | Overall score | Remarks |
|---|---|---|---|---|---|---|---|---|---|---|---|---|
| Williams et al 2020 | Unclear | Unclear | Yes | Yes | Yes | Unclear | No | Yes | Yes | Yes | 6 | Moderate quality |
| Kako et al 2016 | Unclear | Unclear | Yes | Yes | Yes | Yes | Yes | Yes | Yes | Yes | 7 | High quality |
| Burgess and campell 2014 | Unclear | Unclear | Yes | Yes | Yes | No | No | Yes | Yes | Yes | 6 | Moderate quality |

**Table 8. JBI critical appraisal checklist for case reports.**

| AUTHOR'S NAME | YEAR | Were patient's demographic characteristics clearly described? | Was the patient's history clearly described and presented as a | Was the current clinical condition of the patient on presentation clearly | Were diagnostic tests or assessment methods and the results clearly | Was the intervention (s) or treatment procedure(s) clearly described? | We as the post intervention clinical condition clearly described? | Were adverse events (harms) unanticipated events identified and described? | Does the case report provide takeaway lessons? | Overall score | Remarks |
|---|---|---|---|---|---|---|---|---|---|---|---|
| Opiyo et al | 2016 | Yes | Yes | Yes | Yes | Yes | Yes | No | Yes | 7 | High quality |

Ethiopia. In all, 704 FLWHA made up the entire sample size for the three (3) studies. Two studies reported prevalence rates on both anxiety and depression [6, 36].

## 3.4 Mostly used depression and anxiety assessment measures

Among the 19 studies that formally assessed anxiety and depression, 16 (84.2%) used standardized scales/measures while 3 (15.8%) used semi- structured interview guide [37–39]. Among the sixteen (16) studies, five (5) used the Center for Epidemiologic Studies Depression Scale (CES-D) [27, 32, 39–41], four used the Hopkins Symptom Checklist (HSCL-25) [26, 30, 33, 36], three used the Patient Health Questionnaire-9 (PHQ-9) [31, 42], Five studies used different scales thus, the Patient Health Questionnaire-2 (PHQ-2) [29], Beck's Depression Inventory II (BDI-11) [34], Beck's Depression index [35], the Hospital Anxiety and Depression Scale (HADS) [6], and the Emotional distress scale [28]. Most of the scales used were developed and validated in high income settings and therefore lacked contextual (cultural and non -specificity to the HIV population) validity. The reliability coefficient of the scales was not reported in a majority of the studies (n = 11) [6, 26, 29, 30, 32, 33, 35, 37–39, 43].

## 3.5 Risks factors of mental illness in FLWHA

Among the 19 studies included, seventeen (17) studies (89.4%) [6, 25–38, 40, 42] provided data on the risks, correlates and predictors of anxiety and depression among FLWHA in different countries. Regardless of the country, and the type of mental health outcome, these risks factors appear corroboratory. These risk factors arranged in decreasing order include physical intimate partner violence, sexual abuse, HIV/AIDS related stigma, post- traumatic stress disorder, childhood trauma, and higher HIV VL category. Other risk factors included loss of self-esteem, experiences of discrimination, loss of life partners and or other family members, poverty, educational status, victimization, living with people other than direct family members, homelessness, being single or divorced.

In 11 of these studies [6, 26, 28–33, 41–43], a univariate, multivariate or logistic regression analysis showed that sexual abuse, physical intimate partner violence, loneliness, enacted stigma, childhood trauma, post-traumatic stress disorder, young age, high viral load, having no formal education, divorce, unemployment and low social support increased the odds of depression and anxiety in FLWHA and was statistically significant (P ≤ 0.005). However, one study [36] showed there was no significant correlation (P ≥ 0.005) between depression/anxiety symptoms and educational status, employment status, and ART treatment status.

### 3.6 Protective factors of mental health problems in WLWHA

Among the 19 studies included in this review, seven (7) representing 36.8% reported on protective factors against anxiety and depression symptomatology in different countries [6, 27, 32, 34, 36, 38, 43]. Specifically, five (26.3%) of these studies [6, 27, 34, 36, 43] reported on social support as a buffer against anxiety and depression. However, a study conducted by Spies & Seedat [32] in South Africa among FLWHA reported that whiles childhood trauma and posttraumatic stress symptoms contributed to depression severity, resilience reduced the severity of depression, acting as a protective factor. One qualitative study found four coping mechanisms, thus "psychological re-framing of negative situations, mobilization of emotional and financial support from inter-personal networks, churches and HIV support groups" as protective factors [38]. In the same study, participants also sought expert help from "traditional healers, medical services, and/or social workers" less frequently [38].

### 3.7 Impact of HIV/AIDs on mental health of FLWHA

Data on the impact of HIV/AIDS on mental health outcomes of FLWHA was gathered from seven (n = 7) studies representing 36.8% [6, 26, 27, 34, 37, 39, 41] from four countries with relatively different sample characteristics. Among these studies, HIV/AIDS was projected as a significant correlate for depression and anxiety. For example, the study conducted by Spies et al. [41] observed higher rates of depressive symptomatology among women infected with HIV compared to uninfected women. However, three (n = 3) studies, representing 15.8% [31, 38, 40] reported that HIV/AIDS was not an independent driver of anxiety and depression among FLWHA. Two studies also reported that knowledge of HIV/AIDS-positive status and self-disclosure of HIV status was significantly associated with a decrease in depressive symptoms [40, 42].

### 3.8 Impact of mental health on HIV progression

From the included studies for this review, 5 studies [26, 33, 40, 42, 43] reported on the impact of mental health burden on HIV progression among FLWHA. In Sudfeld et al. [33], Tanzania, it was noticed that depression at the start of ART was significantly associated with a higher risk of death (95% CI: 1.15–3.20; P = 0.01) and an increased incidence of severe anaemia (95% CI: 1.07–2.37; P = 0.02) among FLWHA. Again, in the same study, depression was not found to be substantially linked with the trajectory of CD4 T-cell reconstitution or the likelihood of immunologic failure (P value >0.05) but women with depression had increased incidence of oral thrush (HR: 1.15; 0.84–1.58; p = 0.38) and wasting (HR: 1.22; 95% CI: 0.81–1.83; p = 0.34) [33]. In three studies [26, 40, 43], higher levels of depression were linked to a lower likelihood of being virally suppressed. In Govender et al [40], this observation was more likely among younger women (AOR = 0.87, 95% CI [0.78, 0.97], p < 0.05) than older women. Higher score of depression was also highly significantly associated with less utilization of HIV care services and delayed initiation of ART treatment (p < 0.01) [40]

### 4. Discussion

Over the past few decades, there have been an increasing focus on the mental health of women and young girls living with HIV/AIDs particularly in Africa due to their heightened vulnerability to mental illness in the context of HIV [6]. Secondly, the overburdened social roles, cultural, religious and political marginalization of women [4] even in a postmodern philosophical era further threaten the mental health of these vulnerable population. It is not surprising that research and publication on the mental health of FLWHA has gained a momentum boost.

Nonetheless, these research and publications are stacked with individual journals and electronic databases making it difficult for policymakers and relevant stakeholders in the field to have a comprehensive synthesized understanding of the recent developments and evidence regarding mental health burden in FLWHA in Africa.

## 4.1 Prevalence of depression and anxiety

Based on the overall pooled prevalence estimates, the prevalence of depression and anxiety was 5.9% to 61.0% and 28.9% to 61.0% respectively [6, 26, 27, 29–33, 35, 40–42]. Moderately higher levels of depression (45.4%) [44] and anxiety (29.5%) [45] have been reported in the USA and Canada. This reveals the commonalities and realities of mental health issues among this population even across countries with different cultures, geopolitics and socioeconomic indices. That notwithstanding, the literature shows people who are disadvantaged socially and economically are more likely to develop depression [46]. The high prevalence rate in this review emphasizes the need to integrate mental health interventions in the HIV care cascade with special focus on women.

The prevalence rates varied substantially across the individual studies due to the different measurement scales [57], sample size, and geographical context. Even for studies that used the same measurement scale, information on the item number, cut offs and number of included somatic symptoms varied explaining a part of the variability in the data on prevalence rates. Given the earlier concerns of under detection and under diagnosis of depressive and anxiety symptomatology in FLWHA in Africa due to cultural/religious need to suppress the condition rather than reporting and/or clinically manifesting symptoms [15, 47], prevalence rates higher than what is reported in this study is possible. It is crucial that the existing measures/scales for depression and anxiety are modified and assessed for cultural validity. Dessauvagie et al., [48] has argued that scales developed and validated in high-income settings may be inappropriate in diagnosing and quantifying the magnitude of mental health burden in resource-poor settings. The development of a single validated measure for assessing depression and anxiety in sub-Saharan African regions may be a step in the right direction.

## 4.2 Risk factors of depression and anxiety in FLWHA

Our study highlighted a number of intersectional stressors that were considered as the most prevalent and strong predictors of depression and anxiety in the context of HIV. These factors individually or jointly mediated at least a part of the relationship between mental health and HIV. For instance, intimate partner violence (IPV) which affects about 30% of women globally and more prevalent in sub-Saharan Africa (36% of women) [49] were identified by four studies [29–31, 43] as a strong predictor of depression and anxiety in FLWHA. Stigma has shown to negatively impact the quality of life (QoL), social and psychological health and well-being of persons living with HIV [4, 50]. The effect of stigma on QoL is revealed in this review as five studies identified stigma [6, 28, 31, 33, 43] to predict mental health outcomes for FLWHA in Africa. A UNAIDS [50] report uncovered that "pervasive stigma and mental health issues are associated with barriers to access to and engagement in health care". Childhood trauma and posttraumatic stress disorder were implicated in two studies [32, 41] to independently predict depression in FLWHA. The findings in this review align with findings from both low- and high-income countries [51–54]. Collaborative efforts of major stakeholders, health workers, and the community are needed to curb the deleterious mediating effects of these factors on the mental health of FLWHA. Community and healthcare sensitization against stigma [50], the empowerment of women and young girls [55], psychotherapeutic and cognitive behavioural

interventions [4] may be crucial and advantageous at this point as such interventions have proven promising to ultimately impact the mental health outcomes of FLWHA.

## 4.3 Protective factors against depression and anxiety

Social support and resilience may not only have main effects on the mental health outcomes of women living with HIV/AIDS, but may moderate or buffer the negative effects of external and internal stressors on mental health of FLWHA. In one of the studies [6], Pearson's rank correlation showed that social support had a significant negative correlation with depression. Seven studies [6, 27, 32–34, 36, 43] reported on the buffering effects of social support and resilience on depression and anxiety. In addition to their main effects, social support and resilience may function as moderators [56] to decrease the effect of stigma, intimate partner violence, sexual abuse, childhood trauma and posttraumatic stress disorders on the mental health of FLWHA. Therefore, women and young girls living with HIV in Africa may benefit greatly from mental health interventions that focus on building social support and resilience.

## 4.4 Impact of HIV/AIDS on the mental health of FLWHA in Africa

There is enough evidence to suggest that depression is a commonplace for FLWHA and previous research have argued that an HIV diagnosis may exacerbate preexisting mental illness or lead to depression and anxiety symptomatology given the psychosocial distress associated with managing HIV/AIDs [4, 8] especially in Africa. In this current review, HIV/AIDS was projected as the consistent and enduring risk factor for depression and anxiety by seven studies. This is supported by a previous systematic review and meta-analysis [57] that "mental health burden is a logical outcome of HIV/AIDs".

However, contradictory findings on HIV and mental health outcomes emerged in this review. Three studies [31, 38, 40] observed that HIV diagnosis was not an independent predictor of depression and anxiety. Nonetheless, its association with other risk factors are enough to evoke psychological distress among people living with HIV/AIDS [38]. In addition, the review identified that knowledge of HIV positive status was significantly associated with a decrease in depressive symptoms [40, 42]. These divergent outcomes in the evidence is likely because, as discussed previously mental illness in clinical practice is highly underdiagnosed in SSA [15]. Secondly, the lack of a single validated culturally sensitive scale [57] may also contribute to these discrepancies. Lastly, the lack of training for clinicians and the poor integration of mental health in routine HIV care contribute to the under detection and underdiagnosis of depression and anxiety [57]. Albeit, a UNAIDS [50] report also had this to say: "Depression, anxiety and HIV are mutually reinforcing".

## 4.5 Impact of mental health on HIV progression

The presence of depression in FLWHA in the present review was associated with delayed initiation of ART treatment, decreased likelihood of virologic suppression, increased in opportunistic infections, less utilization of HIV care services and higher HIV- related mortality [26, 33, 40, 42, 43]. Previous studies have also established a relationship between mental health and HIV disease progression [4, 56]. It is possible that mental illness may have clinically measurable biological and physiological implication on viral load and physical outcomes of people living with HIV/AIDs [26]. Relationships in the opposite direction is also likely as a few studies have identified ART mediated viral suppression to be associated with a decrease in the severity of depression symptoms [58]. The findings from this review suggest that the lack of screening for and treatment of depression and anxiety among FLWHA [4] has detrimental effects on their HIV related health outcomes. The integration of mental health services into the HIV care

continuum may be an important step towards the achievement of UNAIDS 95-95-95 treatment targets by 2025 and may as well address the bidirectional links between mental health and HIV [59]. Co-integrated mental health services (cognitive behavioural therapy and antidepressants) within HIV clinics have been found in both lower and middle-income countries to significantly improve the timely identification and management of depression and anxiety among HIV-positive individuals [9, 60–62]. This integrated approach reduces the stigma associated with seeking mental health support [62].

## 4.6 Strengths and limitations of this review

This review has several strengths including the use of evidence from 19 studies. Furthermore, the reviewers followed a clear predefined criteria for including and excluding studies. This approach minimized the potential for article selection bias. Also, this review was conducted following well-documented protocols, and the methods used are transparently reported facilitating the reproducibility and replicability to other disciplines. The methodological quality of the included studies was assessed to be of a generally moderate quality allowing for a more objective evaluation of the evidence. The majority of the studies included in this review assessed depression and anxiety using standardized scales.

In spite of these strengths, this review has potential limitations. First, the papers in the final sample cannot represent what has been done in the field of mental health and HIV due to the purposive inclusion criteria used to guide this study. Additionally, our search was limited to 5 databases, and so it is possible that other relevant articles could have been left out. Articles published in languages other than English were excluded, potentially leading to a skewed representation of the evidence. However, they were eliminated because of the cost of translation.

## 4.7 Implications for future interventions and practices

The review's findings highlight the critical need for the development of cost-effective and culturally appropriate mental health therapies for African women living with HIV/AIDS in order to address the specific challenges and disparities they face. Group-based interventions that incorporate culturally relevant practices, such as storytelling or traditional healing, may show promise in promoting mental well-being among African women. Jesse et al. [63] has demonstrated the effectiveness of a culturally tailored group therapy program in alleviating depression and anxiety symptoms among South African women. Task-shifting initiative which involves training non-specialist healthcare workers or community health workers to provide culturally sensitive counseling therapy to African FLWHA within the HIV care can be adopted. In Zimbabwe, the "Friendship Bench Intervention" designed by Chibanda et al. [64] to train lay health workers to deliver evidence-based talk therapy has shown to be cost-effective and feasible in addressing common mental disorders such as depression and anxiety. Co-integrated psychoeducational therapies rooted in the cultural context of African women can improve mental health awareness and coping skills among FLWHA, mitigating the impact of delayed detection and treatment of depression and anxiety.

## 4.8 Conclusion and recommendation

There was high prevalence of depression and anxiety among FLWHA in Africa. This review revealed that the increased burden of mental health issues among this population were mediated by intersectional stressors such as stigma, intimate partner violence and trauma. Social support and resilience appeared to moderate the relationship between HIV/AIDS,

intersectional stressors and mental health burden (depression and anxiety) among FLWHA. A mediation and moderating analysis are needed in the future to authenticate this.

There were contradictory findings on predictive main effect of HIV on mental health burden. While some studies showed an increased in depression and anxiety symptomatology in the face of an HIV diagnosis, others' findings suggested there was no significant relationship between HIV and mental health burden in FLWHA. The lack of existing true experimental findings on HIV and mental health outcomes could account for these discrepancies and the lack of conclusive evidence. Future studies should focus on establishing a causal relationship between HIV and mental illness through experimental studies.

The current evidence on depression and anxiety among FLWHA in sub-Saharan Africa is inadequate and non-widespread as the majority of the studies reporting on this public health menace is concentrated in the South and East of Africa. There also exist substantial methodological limitations in most of the studies. Small sample size, unclear justification of sample size used, and unclear methods of recruitment (sampling methods) resulting in selection bias, a decrease in the statistical power and increased alpha. Future studies should employ standardized methodological approaches, use sufficiently large samples and utilize a robust statistical analysis.

Given the frequency of stresses and mental health issues that FLWH experience, as well as the relationship between the symptoms of mental illness and suboptimal disease treatment, a cost-effective, feasible mental health interventions tailored to the unique and cultural needs of this population are crucial [4].

## Supporting information

**S1 Checklist. PRISMA checklist.**
(DOCX)

**S1 Data. Search strategy for the different databases.**
(DOCX)

## Author Contributions

**Conceptualization:** Dorothy Serwaa Boakye, Mawuko Setordzi.

**Methodology:** Dorothy Serwaa Boakye, Mawuko Setordzi.

**Validation:** Samuel Adjorlolo.

**Writing – original draft:** Dorothy Serwaa Boakye, Mawuko Setordzi.

**Writing – review & editing:** Dorothy Serwaa Boakye, Gladys Dzansi, Samuel Adjorlolo.

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
