## [Decision Letter · Decision Letter 0]

7 Sep 2023

PGPH-D-23-01441

Mental Health Burden among Females Living with HIV and AIDS in sub-Saharan Africa: A systematic review.

Dear Dr. Boakye,

Thank you for submitting your manuscript to PLOS Global Public Health. After careful consideration, we feel that it has merit but does not fully meet PLOS Global Public Health’s publication criteria as it currently stands. Therefore, we invite you to submit a revised version of the manuscript that addresses the points raised during the review process.

We look forward to receiving your revised manuscript.

Kind regards,

Shifa S. Habib

Academic Editor

Journal Requirements:

1. We noticed you have some minor occurrence of overlapping text with the following previous publication(s), which needs to be addressed:

- https://doi.org/10.1080/23311908.2022.2111849

In your revision ensure you cite all your sources (including your own works), and quote or rephrase any duplicated text outside the methods section. Further consideration is dependent on these concerns being addressed.

2. Please provide separate figure files in .tif or .eps format only and remove any figures embedded in your manuscript file. Please also ensure that all files are under our size limit of 10MB. You may leave the figure captions or legends in the manuscript.

3. Please ensure that you refer to Table 1 in your text as, if accepted, production will need this reference to link the reader to the table.

4. Please include a copy of Table 5 which you refer to in your text on page 12.

5. Please include your main tables as part of your main manuscript and remove the individual files.

6. We have noticed that you have uploaded Supporting Information files, but you have not included a list of legends. Please add a full list of legends for your Supporting Information files after the references list.

Additional Editor Comments (if provided):

Reviewers' comments:

Reviewer's Responses to Questions

**Comments to the Author**

1. Does this manuscript meet PLOS Global Public Health’s publication criteria? Is the manuscript technically sound, and do the data support the conclusions? The manuscript must describe methodologically and ethically rigorous research with conclusions that are appropriately drawn based on the data presented.

Reviewer #1: Yes

Reviewer #2: Yes

2. Has the statistical analysis been performed appropriately and rigorously?

Reviewer #1: N/A

Reviewer #2: Yes

3. Have the authors made all data underlying the findings in their manuscript fully available (please refer to the Data Availability Statement at the start of the manuscript PDF file)?

Reviewer #1: No

Reviewer #2: Yes

4. Is the manuscript presented in an intelligible fashion and written in standard English?

Reviewer #1: Yes

Reviewer #2: Yes

5. Review Comments to the Author

Reviewer #1: This manuscript meets PLOS Global Public Health’s publication criteria and has adequate statistical analysis that meets the PLOS Global Public Health’s publication criteria.

The manuscript is intelligibly written, but the authors need to address some minor typographical errors identified and some of these are highlighted in yellow in the manuscript.

Although it was not stated clearly in the article, the authors seemed to use prevalence of depressions and anxiety to indicate the burden of mental health. Although this is acceptable, using other measures such as Quality-Adjusted Life-Years (QALY) and Disability-Adjusted Life-Years (DALY) will have been a strong measure of burden. Did the authors come across articles with QALY and/or DALY measurement and did they try to include such articles? It would have been good if they had gone a step further to look for and use articles with QALY and DALY measurement and include them.

Results: The presentation of the results should conform to the chronology of the objectives.

Some statements appear contradictory and need to be looked at again; e.g. on page 14, two prevalence figures are reported for Tanzania different from the 61% reported from Tanzania. it was not clear if they were still referring to depression or something else. Again, on the same page 5.7% was reported for Nigeria when 5.9% prevalence rate was reported in the earlier sentence. On page 16, the percentage of 7 out of 19 need to be reported accurately; it should be 36.6 and not 38.9%.

Reporting on findings on protective factors needs to be clearer; e.g., while the sentence was almost a direct sentence from the abstract, it is difficult for someone reading it for the first time to identify the four coping mechanisms used by the research participants. It would have been better to write the four coping mechanisms clearer for any reader, and this can be found on page 13-19 of the publication source, specifically on page 13. The authors need to avoid plagiarism and ensure sentences are represented accurately.

Again, on page 17, there was a problem with the representation of the report from the article. The study referenced was conducted in Tanzania and not Uganda and Zambia as mentioned. Secondly, while the key findings of mortality and severe anaemia were measured with statistical significance in the study, these authors did not mention these in their review. They, however, mentioned the one on oral thrush, which was not statistically significant.

The key finding from this study has been highlighted in the manuscript but this is not correct as found in the article. The key finding of this study was “Under the assumption of causality, we estimate 36.1% (95% CI: 13.6-55.1%) of deaths among the study cohort were attributable to depression and its consequences. Depression was not significantly associated with the trajectory of CD4 T-cell reconstitution or the risk of immunologic failure (p-values >0.05).” I recommend this part of the manuscript be rewritten and include the key finding from the article. In addition, the name of the study should be mentioned as it indicates to a large extend to the reader what the study is about.

The discussion section was well-written, but need to be arranged according to the objectives of the review.

Conclusions/recommendations section was well-written. Will the authors consider recommending the development of one common standardised scale/measure for south-Saharan Africa?

The reviewer did not see Table 5.

The attached manuscript has other comments for the attention of the authors.

Reviewer #2: Thank you for the opportunity to review this insightful review.

This work provides a comprehensive review into the mental health challenges faced by females living with HIV/AIDS in sub-Saharan Africa. The study adopts a rigorous systematic review methodology, adhering to the PRISMA guidelines.

The paper opens with a well-structured introduction that clearly establishes the context and significance of the research topic. The description of the research methods is thorough and supported by pertinent information such as search terms and the PRISMA checklist. The review process's transparency is commendable, particularly in the mention of a third reviewer who resolved discrepancies. The study's consideration of quality appraisal scores for each included study enhances its rigor.

However, there are areas for improvement.

• Clarification of Grey Literature Inclusion: While the paper indicates the inclusion of grey literature, it would be beneficial to clarify the handling of these records within the review process. It is unclear whether data from sources beyond the five reported databases were used, and the handling of grey literature needs more clarification, particularly in the PRISMA diagram. I suggest updating the PRISMA diagram to reflect the inclusion of grey literature, thus ensuring a transparent reporting of the methodology. (You can use the PRISMA 2020 flow diagram for new systematic reviews which included searches of databases, registers and other sources).

• Implications for Future Interventions and Practices: The paper could be strengthened by elaborating on how the study's findings can inform future interventions and practices related to mental health issues among FLWHA. While the study's scope may not have encompassed specific solutions, there is room to expand upon potential avenues for addressing mental health challenges. Consider providing more explicit guidance on how mental health interventions could be cost-effective, feasible, and culturally tailored.

• Practical Application: Expanding upon the recommendation for mental health care integration, offer concrete examples or referenced instances where such integration has been successful in similar contexts. This will provide practical insights for mental health service providers and stakeholders working with this population, aligning with the overarching purpose of systematic reviews.

Overall, this paper is a valuable contribution to the understanding of mental health challenges faced by FLWHA in sub-Saharan Africa. It sheds light on the prevalence of depression and anxiety, identifies key correlates and predictors of mental health outcomes, and emphasises the need for culturally tailored interventions. Given the high prevalence rates of mental health issues and their impact on HIV progression, the integration of mental health care into routine HIV care is a logical and essential step.

**Recommendation:** I recommend accepting this paper with minor revisions to address the areas for improvement mentioned above. In summary, the authors should provide more clarity on the inclusion of data sources beyond the five databases reported and the handling of grey literature. Additionally, they should further elaborate on how the findings can inform future interventions and practices, especially in terms of cost-effective and culturally sensitive mental health care for this population in sub-Saharan Africa.

6. PLOS authors have the option to publish the peer review history of their article (what does this mean?). If published, this will include your full peer review and any attached files.

**Do you want your identity to be public for this peer review?** For information about this choice, including consent withdrawal, please see our Privacy Policy.

Reviewer #1: No

Reviewer #2: No

---

## [Decision Letter · Decision Letter 1]

8 Dec 2023

Mental Health Burden among Females Living with HIV and AIDS in sub-Saharan Africa: A systematic review.

PGPH-D-23-01441R1

Dear Ms Boakye,

We are pleased to inform you that your manuscript 'Mental Health Burden among Females Living with HIV and AIDS in sub-Saharan Africa: A systematic review.' has been provisionally accepted for publication in PLOS Global Public Health.

Best regards,

Julia Robinson

Executive Editor

Reviewer Comments (if any, and for reference):

Reviewer's Responses to Questions

**Comments to the Author**

1. If the authors have adequately addressed your comments raised in a previous round of review and you feel that this manuscript is now acceptable for publication, you may indicate that here to bypass the “Comments to the Author” section, enter your conflict of interest statement in the “Confidential to Editor” section, and submit your "Accept" recommendation.

Reviewer #2: All comments have been addressed

2. Does this manuscript meet PLOS Global Public Health’s publication criteria? Is the manuscript technically sound, and do the data support the conclusions? The manuscript must describe methodologically and ethically rigorous research with conclusions that are appropriately drawn based on the data presented.

Reviewer #2: Yes

3. Has the statistical analysis been performed appropriately and rigorously?

Reviewer #2: Yes

4. Have the authors made all data underlying the findings in their manuscript fully available (please refer to the Data Availability Statement at the start of the manuscript PDF file)?

Reviewer #2: Yes

5. Is the manuscript presented in an intelligible fashion and written in standard English?

Reviewer #2: Yes

6. Review Comments to the Author

Reviewer #2: Dear authors.

Thank you for updating your manuscript following my suggestions.

I am happy to see that the manuscript presents the methodology in a more readable way and that you added a section on the implications for future interventions and practices. I truly believe that this will help future readers to understand your work and use it for prospective projects.

Congratulations.

7. PLOS authors have the option to publish the peer review history of their article (what does this mean?). If published, this will include your full peer review and any attached files.

**Do you want your identity to be public for this peer review?** For information about this choice, including consent withdrawal, please see our Privacy Policy.

Reviewer #2: **Yes: **Germán Andrés Alarcón Garavito
